# UniGADD: Universal Generator-Agnostic Deepfake Detector

## Abstract

In this paper, we introduce Universal Generator-Agnostic Deepfake Detector (UniGADD), a novel method that employs supervised learning to achieve high accuracy on known deepfake generators, while maintaining robust performance on previously unseen ones. The proposed approach follows a two-stage optimisation process. In the first stage, a contrastive loss encourages the model to learn discriminative feature embeddings from real and known fake images, resulting in strong performance within the training domain. In the second stage, the embedding space is refined by promoting inter-cluster separation and intra-cluster compactness, applied exclusively to real samples. This refinement enhances generalisability, enabling the method to exhibit improved robustness against unseen deepfake generation techniques. UniGADD achieves accuracy on par with state-of-the-art methods for known generators, while significantly outperforming them on unseen cases, demonstrating its scalability and practical applicability for adversarial content detection.

## 1 Introduction

The rapid proliferation of deepfake generators poses an escalating challenge for forensic tools (Rana et al., 2022). Most published detectors are trained with labelled examples from a fixed set of generators, achieving high accuracy on those seen attacks yet faltering when confronted with novel manipulation pipelines. Pellicer *et al.* (Pellicer et al., 2024) frame the task as unsupervised anomaly detection, showing that generator-agnostic cues exist, although their reliance on nominal-only statistics leaves recent advances in representation learning under-explored.

To address the cold-start scenario where no fake examples are available, we refine foundation-model embeddings with a dedicated contrastive loss and derive prototypes that map the real-image distribution. Samples that deviate from these prototypes are flagged as potential anomalies, enabling detection without generator-specific supervision. This approach sidesteps the need for curated fake datasets and remains robust as synthesis methods evolve.

While such unsupervised strategies excel in generalisability, purely nominal training can sacrifice closed-set precision. Conversely, supervised detectors capture fine-grained artefacts but overfit to the generators present in the training set (Wang et al., 2024). We propose the **Universal Generator-Agnostic Deepfake Detector (UniGADD)**, which unifies these complementary strengths. UniGADD first exploits labelled fakes, when available, to carve a discriminative margin between real and manipulated imagery, then tightens the real-image region through the prototype-based refinement introduced above. The result is a detector that preserves high precision on known attacks while degrading gracefully on unseen ones.

Two practical considerations guide our design. Authentic imagery is plentiful, yet collecting labelled samples for every new generator is unrealistic (Naitali et al., 2023). At the same time, detectors must remain reliable as synthesis technology evolves. By combining light supervision with feature-space regularisation, UniGADD meets both requirements without continual retraining.

We benchmark UniGADD on the AIGC datasets (Fan et al., 2024), which contain a diverse mixture of GAN- and diffusion-based generators and reflect varied real-world usage scenarios (Zhu et al., 2023). UniGADD matches or surpasses specialist baselines on their home domains and maintains competitive performance when confronted with previously unseen generators, supporting its suitability for real-world deployment.

Our main contributions are as follows:

- We introduce UniGADD, a contrastive clustering framework that generalizes synthetic image detection to unseen generative models.

- We propose a real-image clustering strategy that enables the model to leverage intrinsic data structure without supervision.

- We establish a leave-one-out protocol covering both GAN- and diffusion-based generators, enabling rigorous evaluation of cross-domain transfer.

- We demonstrate competitive performance on the challenging AIGC benchmark, highlighting strong one-to-many generalization.

**Terminology.** Throughout this paper we use the term "deepfake" in a broad sense to mean AI-generated visual content (AIGC). We adopt this wording because "deepfake" has become the prevalent public term for generated images, but whenever precise distinctions are required we explicitly refer to GAN- or diffusion-based synthesis.

## 2 Related Work

Early forensic efforts relied on handcrafted cues such as lighting, facial landmarks, or head pose (Farid, 2016; Sencar & Memon, 2013). As GANs and diffusion models advanced, supervised classifiers trained on curated real/fake datasets became standard (Rössler et al., 2019; Yang et al., 2019; Wang et al., 2020a; Bird & Lotfi, 2023), with DE-FAKE (Sha et al., 2023) and the contrastive method of Baraldi *et al.* (Baraldi et al., 2024) demonstrating strong closed-set accuracy. These systems, however, tend to generalise poorly to unseen generators, as documented by Ojha *et al.* (Ojha et al., 2023). Frequency-domain studies report similar limitations, with artefacts learnt from GAN families rarely transferring to diffusion models (Frank et al., 2020; Wang et al., 2020b).

Generator-agnostic detection aims to mitigate this brittleness. Epstein *et al.* (Epstein et al., 2023) show substantial accuracy drops when a novel generator appears in a continual-learning setting, and Corvi *et al.* (Corvi et al., 2023a;b) confirm that detectors trained on GAN artefacts do not automatically extend to diffusion artefacts. Prototype-based detectors, exemplified by PUDD (Pellicer et al., 2024), measure similarity to learned class prototypes and flag low-score inputs as potential fakes, improving robustness to unseen attacks. Recent work couples contrastive objectives with large vision encoders to capture both discrimination and generalisation: Larue *et al.* (Larue et al., 2023) compact the real cluster with a bounded loss, Sun *et al.* (Sun et al., 2023) use pseudo-labels for open-world attribution, and Shi *et al.* (Shi et al., 2023) report that linear probes on CLIP features can outperform many bespoke detectors, although fine-tuning still offers gains.

Visual anomaly-detection methods model the nominal data distribution and treat deviations as outliers. PatchCore (Roth et al., 2022) stores patch features in a memory bank, and PaDiM (Defard et al., 2020) employs pixel-wise Gaussian models. While effective in industrial settings, these designs entail memory overhead or pixel-level supervision that does not transfer cleanly to deepfake tasks. Foundation-model embeddings from CLIP (Radford et al., 2021b) and DINOv2 (Oquab et al., 2024) provide a strong baseline, yet adapting them without sacrificing general utility remains challenging; Han *et al.* (Han et al., 2024) show that fine-tuned variants can overfit the target domain.

Prototypical networks compute class centroids in the embedding space and classify samples by distance (Snell et al., 2017). Their success in few-shot learning and interpretability has motivated extensions to meta-learning and self-supervised representation learning (Chen et al., 2018; Angelov et al., 2025; Donnelly et al., 2021; Wu et al., 2021). SwAV (Caron et al., 2020) and DINO (Caron et al., 2021) organise unlabelled data into semantically meaningful clusters, although prototype collapse can limit diversity, prompting regularisation strategies such as ReSA (Weng et al., 2025). The contrastive loss used in UniGADD builds on this body of work by enforcing inter-prototype separation and intra-prototype compactness within the real-image

manifold, enabling efficient anomaly cues while retaining the general features of the underlying foundation model.

UniGADD integrates supervised contrastive learning, prototype-based refinement, and lightweight anomaly detection, preserving high precision on generators seen during training and improves recall on unseen ones without requiring extensive memory resources or continual retraining.

# 3 Proposed Method

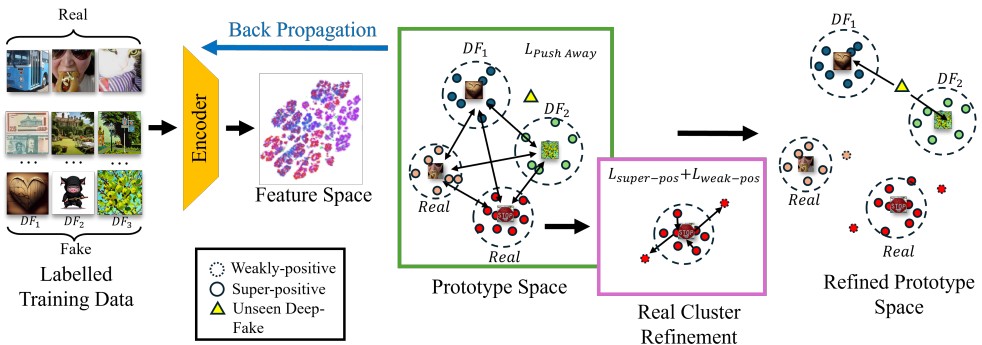

Figure 1: Overview of the proposed method, consisting of feature extraction, supervised contrastive prototype refinement and anomaly detection.

We propose a supervised contrastive learning framework for deepfake detection and generator attribution that combines structured pseudo-labelling of real data with explicit supervision from known fake generators. UniGADD follows a two-stage process: the first learns a discriminative embedding space through hybrid labelling, while the second refines cluster structure within the real image distribution to improve robustness and generalisability.

## 3.1 Feature Extraction and Prototype Initialisation

Given a set of real and fake training samples $X = \{\mathbf{x}_i\}_{i=1}^{N}$, we extract feature embeddings using state-of-the-art Vision Transformer architectures, including DINOv2 (Oquab et al., 2024) and CLIP (Radford et al., 2021a). DINOv2 produces 1024-dimensional embeddings, while CLIP generates 768-dimensional representations. These embeddings define a unified representation space for both real and fake data, providing a solid foundation for contrastive learning and structural analysis.

We begin by applying K-means clustering over the real sample embeddings to induce an initial structure in the feature space. These cluster assignments serve as pseudo-labels for real samples, while fake samples retain their known generator labels. The combined labels are used to supervise contrastive learning, which sharpens the representation structure by encouraging alignment within clusters and separation between them. Crucially, this sharpening lays the foundation for the second stage, where we explicitly refine the real clusters through intra-cluster compactness objectives. After contrastive training, we perform prototype selection by identifying the samples closest to each cluster centre in the refined embedding space. This deferred selection ensures that the prototypes reflect the improved geometry of the learned space, leading to enhanced interpretability and robustness in downstream detection and attribution.

## 3.2 Stage 1: Supervised Contrastive Learning

To learn a discriminative embedding space, we optimise a supervised contrastive loss (Khosla et al., 2020) over a composite label set. Let $\mathbf{z}_i$ denote the normalised embedding of sample $i$, and let $y_i$ be its composite

label. The loss is defined as:

$$L_{\text{contrast}} = \sum_{i=1}^{N} \frac{-1}{|\mathcal{P}(i)|} \sum_{p \in \mathcal{P}(i)} \log \frac{\exp(\mathbf{z}_i \cdot \mathbf{z}_p / \tau)}{\sum_{a \in \mathcal{A}(i)} \exp(\mathbf{z}_i \cdot \mathbf{z}_a / \tau)}, \tag{1}$$

where $\mathcal{P}(i) = \{j \neq i \,|\, y_j = y_i\}$ denotes the set of positives for anchor $i$, $\mathcal{A}(i) = \{j \neq i\}$ is the full batch excluding the anchor, and $\tau$ is a temperature parameter.

This objective encourages embeddings of samples with the same composite label, either real samples sharing a pseudo-cluster label or fake samples from the same generator, to be drawn together, while pushing apart dissimilar instances. Crucially, the pseudo-labels for real samples, derived from K-means clustering, are periodically updated during training to reflect the evolving geometry of the feature space. This dynamic relabelling reinforces the underlying structure and helps sharpen cluster boundaries, setting the stage for more effective intra-cluster refinement.

### 3.3 Stage 2: Intra-Cluster Compactness

Following supervised contrastive learning, we further refine the structure of the real sample distribution by explicitly enforcing intra-cluster compactness. The pseudo-clusters induced in Stage 1 provide an initial structural prior over the real data. However, these clusters may contain internal variation, and loosely aligned samples near the periphery can obscure clear decision boundaries, especially when facing previously unseen deepfakes.

To address this, we define a two-part objective that consolidates cluster structure by tightening the core and softly regulating the boundary. For each real cluster $k$, we compute a prototype vector $\boldsymbol{\mu}_k$ as the mean embedding of the assigned samples. Cosine similarity is then used to assess alignment between a sample and its cluster centre:

$$s_k(\mathbf{z}_i) = \frac{\mathbf{z}_i \cdot \boldsymbol{\mu}_k}{\|\mathbf{z}_i\| \|\boldsymbol{\mu}_k\|}, \tag{2}$$

$$X_k^{(sp)} = \{\mathbf{z}_i : y_i = k, \ s_k(\mathbf{z}_i) \geq \mu_k - 2\sigma_k\}, \tag{3}$$

$$X_k^{(wp)} = \{\mathbf{z}_i : y_i = k, \ s_k(\mathbf{z}_i) < \mu_k - 2\sigma_k\}, \tag{4}$$

where $\mu_k$ and $\sigma_k$ denote the mean and standard deviation of similarity scores within cluster $k$.

The super-positive region $X_k^{(sp)}$ represents highly aligned, clean real samples. By pulling them closer to the prototype, we reinforce cluster cohesion:

$$L_{\text{super-pos}} = \sum_{k=1}^{K_r} \sum_{\mathbf{z}_i \in X_k^{(sp)}} \left(1 - s_k(\mathbf{z}_i)\right)^2. \tag{5}$$

Conversely, the weak-positive region $X_k^{(wp)}$ captures samples that deviate from the cluster core. These peripheral areas are likely to overlap with novel or unseen fake samples in the test phase. To improve anomaly sensitivity, we gently push such samples outward using:

$$L_{\text{weak-pos}} = \sum_{k=1}^{K_r} \sum_{\mathbf{z}_i \in X_k^{(wp)}} s_k(\mathbf{z}_i). \tag{6}$$

The total loss is given by:

$$L_{\text{refine}} = \lambda_1 L_{\text{super-pos}} + \lambda_2 L_{\text{weak-pos}}, \tag{7}$$

where $\lambda_1$ and $\lambda_2$ control the relative contribution of each term.

This refinement stage not only improves the clarity and robustness of real clusters but also increases the likelihood that unseen fake samples will fall outside the compacted real distribution, enabling more accurate detection and attribution.

### 3.4 Deepfake Detection and Attribution

Following the two-stage training process, we compute final cluster centroids for both real and fake classes. These centroids summarise the distributional structure of the training data in the embedding space, where real clusters have been compacted through refinement, and fake clusters are formed based on known generator groupings established during contrastive learning.

For a test embedding $\mathbf{z}_{\text{test}}$, we calculate its Euclidean distance to each cluster centroid. The sample is classified as real if it is closer to any real cluster than to the closest fake cluster:

$$\text{label} = \begin{cases} \text{Real}, & \text{if } \min_k \|\mathbf{z}_{\text{test}} - \boldsymbol{\mu}_k^{\text{real}}\| < \min_j \|\mathbf{z}_{\text{test}} - \boldsymbol{\mu}_j^{\text{fake}}\|, \\ \text{Fake}_{j^*}, & \text{otherwise, with } j^* = \arg\min_j \|\mathbf{z}_{\text{test}} - \boldsymbol{\mu}_j^{\text{fake}}\|. \end{cases} \tag{8}$$

This nearest-centroid classification mechanism enables simultaneous deepfake detection and generator attribution. The refined structure of real clusters ensures a compact decision region, increasing sensitivity to out-of-distribution manipulations. Meanwhile, fake centroids, which are anchored by generator-specific labels, allow for accurate attribution among known manipulations.

Importantly, since weakly aligned real samples have been softly pushed away from the cluster cores, previously unseen fake content is more likely to fall outside the refined real distribution. This results in improved robustness against generalisation failures and supports open-set detection capabilities in realistic forensic settings.

---

**Algorithm 1:** Supervised Contrastive Detection Framework

---

**Input:** Embeddings, ground-truth generator labels, number of real clusters $K_r$, learning rate $\eta$

`// Stage 1:  Contrastive embedding learning`

**for** $epoch = 1$ *to #Stage 1 epochs* **do**

  Cluster real samples using k-means

  Assign composite labels (real pseudo-labels + fake generator labels)

  Compute contrastive loss $L_{\text{contrast}}$

  Update projection model using gradient descent

**end**

`// Stage 2:  Cluster refinement on real data`

**for** $epoch = 1$ *to #Stage 2 epochs* **do**

  Compute real cluster prototypes $\boldsymbol{\mu}_k$

  Identify $X_k^{(sp)}$, $X_k^{(wp)}$ via cosine similarity

  Compute $L_{\text{super-pos}}$, $L_{\text{weak-pos}}$

  Update model using gradient descent

**end**

`// Inference and attribution`

Store final centroids for real and fake clusters

Classify test samples by nearest-centroid comparison

---

**Visualising Embedding Structure.** To qualitatively assess the structure induced by our training procedure, we project the learned embeddings into 2D using t-SNE. Figure 2 shows the resulting layouts when individual generators are held out during training. Real samples form compact, well-separated clusters, while fake samples from unseen generators appear peripheral or distinct, illustrating UniGADD's ability to isolate anomalous inputs in feature space.

## 4 Experiments

**Implementation details.** We evaluate UniGADD on the AIGC datasets (Fan et al., 2024), covering diverse synthesis styles and content types. Images are resized to $224 \times 224$ and normalised with ImageNet statistics. Features are extracted using a frozen DINOv2 ViT-L/14 backbone (Oquab et al., 2024), and a

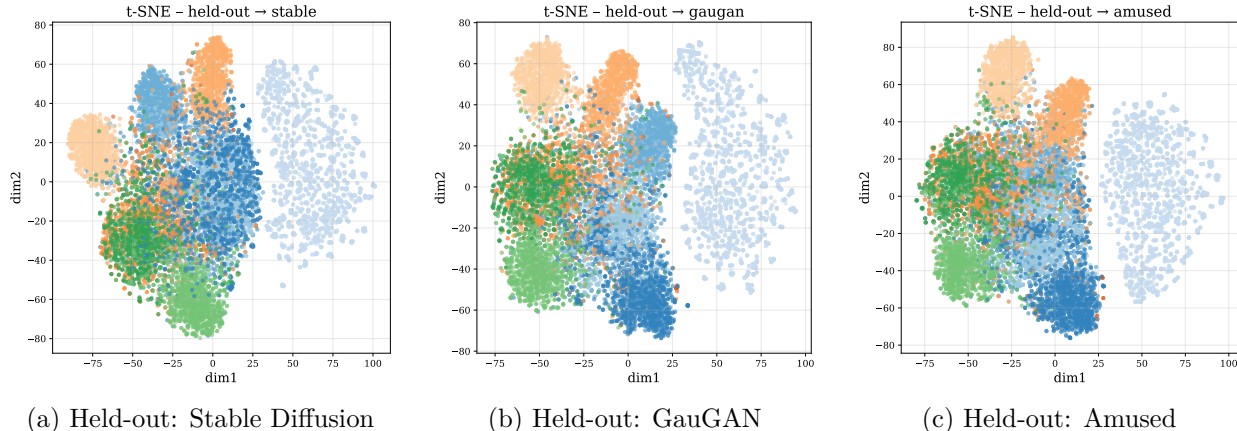

(a) Held-out: Stable Diffusion          (b) Held-out: GauGAN          (c) Held-out: Amused

Figure 2: t-SNE projections of the embedding space after training. Light blue indicates the real samples. Real samples form compact clusters, while unseen generator samples (held out during training) remain distinct or peripheral.

linear projection head maps embeddings to 512 dimensions. The projection is trained for 10 epochs with Adam ($\text{lr} = 10^{-3}$), contrastive batch size 256, and temperature $\tau = 0.07$.

Real images are partitioned into clusters to provide pseudo-labels during training. We fix $n_{\text{total}} = 40$ clusters, with $n_{\text{real}} = n_{\text{total}} - n_{\text{fake}}$ where $n_{\text{fake}}$ is the number of training generators. After training, class centroids are computed in the learned space and used for nearest-centroid classification at test time. The total number of clusters controls the granularity of the real-image manifold in the embedding space. A smaller number of clusters yields coarse partitions that may under-represent intra-class variation, while a larger number of clusters produces finer partitions that better capture local structure. In our setting, increasing the number of real clusters allows the model to more tightly characterise the nominal data distribution, which in turn improves separation at the boundary where unseen fake samples are likely to appear. Empirically, we observe that performance remains stable across a wide range of cluster counts (see Table 4), indicating that the method is not overl y sensitive to this hyperparameter. We therefore select a moderate value that balances representational fidelity and computational cost.

Evaluation follows a leave-one-out protocol: each generator is excluded in turn and used as an unseen test set, while the remaining generators and all real data form the training pool. Splits are sampled with a fixed random seed for reproducibility. All experiments are implemented in PyTorch and run on NVIDIA Tesla V100 GPUs.

## 5 Results

### 5.1 Evaluation Protocol

We report results using class-averaged accuracy (%) to ensure equal weighting across real and fake classes. To evaluate generalisation under distribution shift, we separate test data into *seen* (generators included during training) and *unseen* (generators excluded) subsets. This separation simulates realistic deployment conditions where previously unknown generation techniques may emerge.

We compare UniGADD against recent state-of-the-art deepfake detection methods, including supervised and unsupervised approaches (Ojha et al., 2023; Wang et al., 2023; Cozzolino et al., 2024). We also benchmark a DINOv2 + Linear classifier baseline to demonstrate that our method improves upon standard frozen-feature detectors by incorporating contrastive structure and cluster refinement. Our experimental design progressively increases the challenge of generalisation. We begin with a Leave-One-Out (LOO) setup, withholding one generator at a time, then extend to a Leave-$N$-Out (LNO) evaluation where multiple generators are

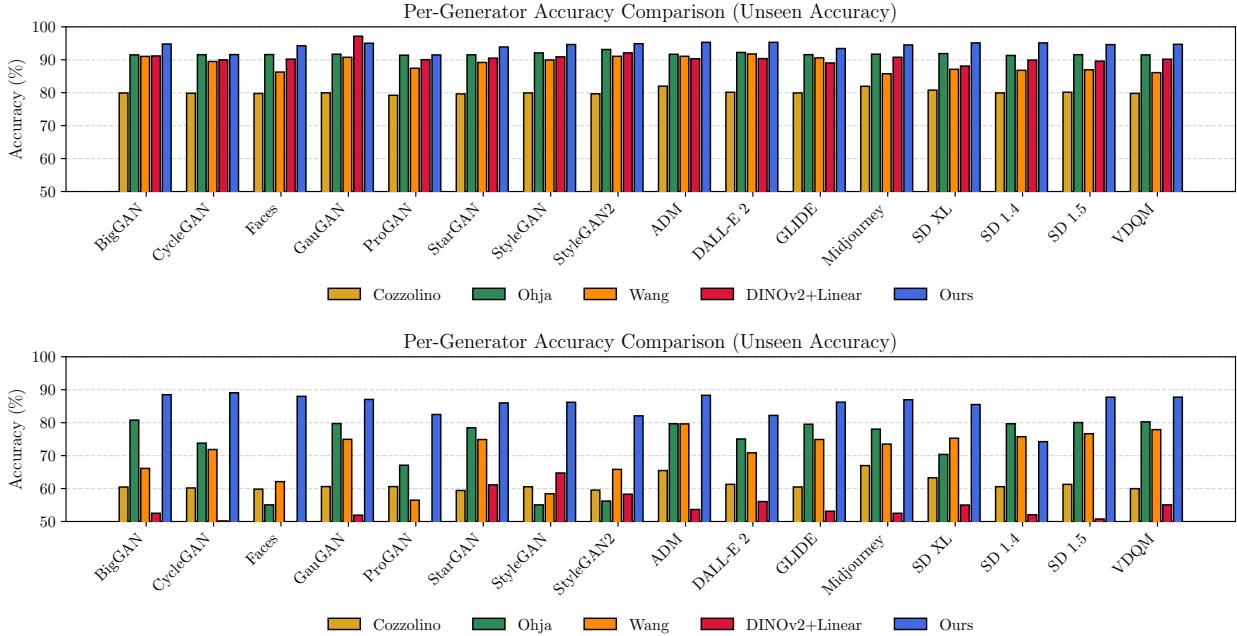

Figure 3: Average accuracy per generator under leave-one-out evaluation. Subfigure (a) shows accuracy on seen generators; (b) shows accuracy on unseen ones. UniGADD maintains high performance across both, outperforming previous methods in generalisation.

excluded simultaneously. Finally, we examine the effect of changing the backbone encoder to test whether UniGADD's robustness is architecture-dependent.

| | Method → | Cozzolino et al. ↑ | | Ohja et al. ↑ | | Wang et al. ↑ | | DINOv2 + Linear ↑ | | Ours ↑ | |
|---|---|---|---|---|---|---|---|---|---|---|---|
| | ↓ Generator | *Seen* | *Unseen* | *Seen* | *Unseen* | *Seen* | *Unseen* | *Seen* | *Unseen* | *Seen* | *Unseen* |
| **GANs** | BigGAN | 79.92 | 60.47 | 91.49 | 80.79 | 91.03 | 66.12 | 91.16 | 52.50 | 94.80 | 88.50 |
| | CycleGAN | 79.84 | 60.18 | 91.55 | 73.79 | 89.48 | 71.86 | 89.97 | 50.17 | 91.58 | 89.10 |
| | Faces | 79.77 | 59.82 | 91.59 | 55.05 | 86.27 | 62.10 | 90.20 | 48.04 | 94.24 | 88.00 |
| | GauGAN | 79.96 | 60.60 | 91.69 | 79.74 | 90.76 | 74.95 | 97.16 | 51.92 | 95.03 | 87.06 |
| | ProGAN | 79.22 | 60.60 | 91.39 | 67.09 | 87.42 | 56.47 | 90.04 | 45.06 | 91.47 | 82.50 |
| | StarGAN | 79.64 | 59.42 | 91.51 | 78.47 | 89.19 | 74.88 | 90.50 | 61.12 | 93.88 | 86.00 |
| | StyleGAN | 79.93 | 60.53 | 92.09 | 55.05 | 89.94 | 58.41 | 90.89 | 64.73 | 94.64 | 86.20 |
| | StyleGAN2 | 79.67 | 59.53 | 93.13 | 56.21 | 91.06 | 65.83 | 92.10 | 58.29 | 94.87 | 82.08 |
| **Diffusion** | ADM | 81.98 | 65.47 | 91.67 | 79.68 | 91.06 | 79.63 | 90.32 | 53.64 | 95.30 | 88.32 |
| | DALL-E 2 | 80.14 | 61.28 | 92.23 | 75.05 | 91.77 | 70.85 | 90.37 | 56.06 | 95.30 | 82.21 |
| | GLIDE | 79.93 | 60.48 | 91.55 | 79.53 | 90.58 | 74.90 | 89.03 | 53.12 | 93.42 | 86.23 |
| | Midjourney | 81.95 | 66.98 | 91.69 | 78.05 | 85.75 | 73.50 | 90.77 | 52.50 | 94.51 | 86.97 |
| | SD XL | 80.78 | 63.27 | 91.88 | 70.37 | 87.10 | 75.28 | 88.10 | 55.00 | 95.13 | 85.51 |
| | SD 1.4 | 79.94 | 60.57 | 91.32 | 79.68 | 86.80 | 75.74 | 89.91 | 52.06 | 95.13 | 74.25 |
| | SD 1.5 | 80.15 | 61.28 | 91.55 | 80.05 | 86.96 | 76.64 | 89.58 | 50.75 | 94.61 | 87.75 |
| | VDQM | 79.80 | 59.98 | 91.49 | 80.26 | 86.10 | 77.90 | 90.17 | 55.05 | 94.70 | 87.75 |
| | **AVG** | 80.16 | 59.82 | 91.68 | 73.64 | 88.62 | 70.79 | 90.32 | 53.84 | 94.29 | 85.53 |
| | **Range ↓** | 2.76 | 22.50 | 0.91 | 25.74 | 5.68 | 21.43 | 9.06 | 19.67 | 3.83 | 14.85 |

Table 1: Detection performance by generator across different methods for Leave-One-Out, showing *Seen* and *Unseen* accuracy. Generators are grouped by model type (GANs vs Diffusion). Our (UniGADD) results are highlighted in green.

## 5.2 Leave-One-Out Generalisation

We first assess UniGADD's performance in the LOO setting. In each leave-one-out run, one generator is held out in turn, ensuring that every dataset serves once as the unseen test domain; within each split, train/test partitions are sampled randomly on multiple runs and averaged. Results are presented in Table 1 and visualised in Figure 3. It should be noted that the held-out sets include both GAN-based and diffusion-based models (e.g., excluding GauGAN while training on diffusion models, or vice versa), our protocol naturally covers cross-paradigm transfer between GANs and diffusion models.

Across all methods, performance on unseen generators is consistently lower than on seen ones, confirming the difficulty of out-of-distribution detection. However, UniGADD achieves the best unseen accuracy overall (**85.53%**), significantly outperforming prior methods, including supervised (Ohja: 73.64%) and unsupervised baselines (DINOv2+Linear: 53.84%). UniGADD also exhibits the lowest variance across unseen generators (**range: 14.85**), suggesting consistent robustness even when exposed to unfamiliar generation styles. Notably, diffusion models—often harder to detect—see substantial improvements with UniGADD. For example, on Midjourney and ADM, our method achieves over **86%** accuracy, whereas DINOv2+Linear drops below 55%. These findings validate that the contrastive-prototype training and real-cluster refinement stages equip UniGADD with stronger resilience to distribution shift than previous approaches.

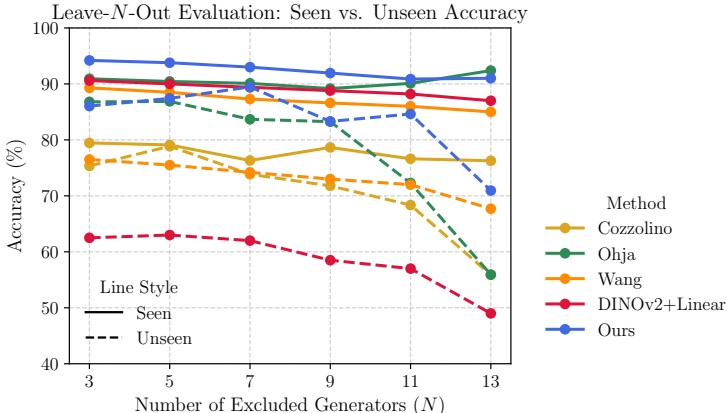

Figure 4: Average accuracy as the number of generators excluded from training increases. UniGADD generalises more robustly than prior methods.

### 5.3 Leave-$N$-Out Generalisation

To further evaluate generalisation under increasingly challenging scenarios, we conduct a Leave-$N$-Out evaluation, varying $N$ from 3 to 13. For each $N$, 10 random generator subsets are withheld from training and averaged to ensure stability. Results are summarised in Table 2 and visualised in Figure 4.

As expected, all methods experience performance degradation as $N$ increases. However, UniGADD retains high accuracy on both seen (**92.47%**) and unseen generators (**83.63%** on average). Even with 13 generators excluded—nearly 80% of all available training data—UniGADD outperforms others by large margins. Compared to Ohja et al., our method shows a smaller drop in unseen accuracy from $N = 3$ to $N = 13$ (**15.09pp vs 30.86pp**), underscoring UniGADD's graceful degradation under extreme distributional shifts. The range for unseen accuracy is also lowest for UniGADD (**18.51**), again demonstrating stability across permutations. This experiment supports our hypothesis that refining the geometry of the real distribution during training leads to broader generalisation across diverse, previously unobserved generative mechanisms.

| Method → | Cozzolino et al. ↑ | | Ohja et al. ↑ | | Wang et al. ↑ | | DINOv2 + Linear ↑ | | Ours ↑ | |
|---|---|---|---|---|---|---|---|---|---|---|
| ↓ $N$ **Unseen** | *Seen* | *Unseen* | *Seen* | *Unseen* | *Seen* | *Unseen* | *Seen* | *Unseen* | *Seen* | *Unseen* |
| 3 | 79.455 | 75.35 | 90.92 | 86.79 | 89.3 | 76.5 | 90.6 | 62.5 | 94.20 | 86.04 |
| 5 | 79.10 | 78.86 | 90.44 | 86.88 | 88.5 | 75.5 | 90.0 | 63.0 | 93.79 | 87.42 |
| 7 | 76.32 | 73.86 | 90.10 | 83.68 | 87.3 | 74.2 | 89.4 | 62.0 | 93.00 | 89.46 |
| 9 | 78.65 | 71.79 | 89.17 | 83.26 | 86.6 | 73.0 | 88.8 | 58.5 | 91.94 | 83.30 |
| 11 | 76.62 | 68.35 | 90.10 | 72.29 | 86.0 | 72.0 | 88.2 | 57.0 | 90.87 | 84.63 |
| 13 | 76.27 | 55.90 | 92.39 | 55.93 | 85.0 | 67.7 | 87.0 | 49.0 | 91.01 | 70.95 |
| **AVG** | 77.74 | 73.73 | 90.52 | 78.14 | 87.12 | 73.15 | 89.00 | 58.67 | 92.47 | 83.63 |
| **Range ↓** | 3.19 | 29.20 | 3.22 | 30.95 | 4.30 | 8.80 | 3.60 | 14.00 | 3.33 | 18.51 |

Table 2: Detection performance by generator across different methods for Leave-$N$-Out, showing *Seen* and *Unseen* accuracy. Our (UniGADD) results are highlighted in green.

## 5.4 Backbone Robustness Analysis

To ensure our results are not backbone-specific, we perform ablations using four different pretrained vision encoders: CLIP ViT-B/32, CLIP ViT-L/14, DINOv2-S/14, and DINOv2-B/14 (our default). Table 3 reports seen and unseen accuracy for each configuration under the LOO setting.

CLIP-based backbones, while strong on seen data (up to **96.72%**), exhibit larger gaps on unseen samples, particularly for the smaller ViT-B/32 (only **70.57%** unseen accuracy). In contrast, DINOv2-B/14 achieves better balance with **92.48%** seen and **81.47%** unseen accuracy, offering a compelling trade-off between closed- and open-set performance.

These results demonstrate that while UniGADD benefits from stronger backbone representations, its core generalisation ability holds across architectures. The refinement stage appears to amplify the strengths of foundation models while mitigating their overfitting tendencies. Further ablation studies analysing the sensitivity of UniGADD to clustering configuration, decision thresholds, backbone choice, and loss formulation are provided in Appendix A.

| | Backbone → | ViT-B-32 ↑ | | ViT-L-14 ↑ | | DINOv2-B/14 ↑ | | DINOv2-S/14 ↑ | |
|---|---|---|---|---|---|---|---|---|---|
| | ↓ **Generator** | *Seen* | *Unseen* | *Seen* | *Unseen* | *Seen* | *Unseen* | *Seen* | *Unseen* |
| **GANs** | BigGAN | 89.54 | 62.00 | 96.55 | 88.50 | 91.02 | 91.25 | 81.91 | 65.75 |
| | CycleGAN | 87.57 | 71.50 | 96.68 | 88.50 | 92.83 | 84.25 | 79.93 | 69.50 |
| | Faces | 86.74 | 57.25 | 96.68 | 86.50 | 89.70 | 76.25 | 79.11 | 63.25 |
| | GauGAN | 90.72 | 67.75 | 96.38 | 83.50 | 91.41 | 87.25 | 82.47 | 68.25 |
| | ProGAN | 89.12 | 76.25 | 96.91 | 81.25 | 94.22 | 77.92 | 81.56 | 76.25 |
| | StarGAN | 87.96 | 66.25 | 96.88 | 88.25 | 90.20 | 89.50 | 80.10 | 78.50 |
| | StyleGAN | 89.80 | 62.75 | 97.01 | 81.25 | 94.24 | 70.00 | 82.30 | 58.00 |
| | StyleGAN2 | 87.07 | 63.75 | 96.32 | 87.75 | 94.54 | 81.75 | 78.98 | 61.00 |
| **Diffusion** | ADM | 89.80 | 70.00 | 96.97 | 87.75 | 93.12 | 87.75 | 81.25 | 66.00 |
| | DALL-E 2 | 89.21 | 70.50 | 96.58 | 87.50 | 93.06 | 77.75 | 86.25 | 74.25 |
| | GLIDE | 87.60 | 76.25 | 96.74 | 88.25 | 89.51 | 74.25 | 83.29 | 59.50 |
| | Midjourney | 92.57 | 78.00 | 97.24 | 87.25 | 93.88 | 81.50 | 85.10 | 76.50 |
| | SD XL | 92.73 | 80.25 | 96.84 | 87.00 | 91.68 | 79.25 | 81.55 | 72.50 |
| | SD 1.4 | 88.82 | 80.25 | 96.78 | 87.00 | 94.38 | 86.25 | 83.85 | 79.25 |
| | SD 1.5 | 90.26 | 81.25 | 96.64 | 84.75 | 92.57 | 83.25 | 82.17 | 74.50 |
| | VDQM | 89.84 | 69.25 | 96.81 | 85.50 | 92.96 | 86.50 | 82.83 | 67.00 |
| | **AVG** | 89.15 | 70.57 | 96.72 | 86.53 | 92.48 | 81.47 | 81.86 | 68.56 |
| | **Range ↓** | 6.00 | 24.00 | 1.69 | 7.25 | 5.03 | 21.25 | 7.29 | 21.50 |

Table 3: Detection accuracy by backbone across GAN and Diffusion generators under Leave-One-Out setting.

## 5.5 Limitations

UniGADD shows strong generalisation to unseen generators, but it has several limitations. The method relies on frozen vision transformer backbones such as DINOv2 and CLIP, so any biases or weaknesses in these models are inherited. For example, detection may be less reliable under unusual lighting conditions or non-photographic image styles. The current method only uses single images and does not capture temporal signals that can be useful for spotting subtle video forgeries. Extending UniGADD to video is an important direction for future work.

Our experiments are based on balanced datasets with a reasonable number of samples per generator. The performance of UniGADD in low-shot settings, highly imbalanced data, or in-the-wild content remains untested. Although inference is fast and lightweight, the clustering step used for prototype refinement becomes more expensive as the dataset grows. Large-scale deployments may require streaming or incremental clustering methods.

We do not explicitly evaluate robustness to low-level image perturbations such as compression or blur, which remain important directions for extending our approach.

Finally, we use a fixed threshold for classification, selected on a held-out validation set. In practical deployments, the decision boundary may need to adapt over time as content distributions shift. These points highlight areas where further refinement is needed before UniGADD is suitable for production-level deployment.

# 6 Discussion & Conclusion

We introduced UniGADD, a generator-agnostic deepfake detector that performs well on both seen and unseen manipulations. The method uses a two-stage approach. First, supervised contrastive learning builds a structured embedding space using real and labelled fake examples. Then, a refinement step improves the compactness of real-image clusters, making it easier to detect outliers caused by new manipulations. Results show that UniGADD matches the accuracy of other detectors on known generators and outperforms them when generalising to novel ones. It maintains high accuracy even as more generators are excluded from training. The method also shows low variance across different settings, which suggests consistent performance.

We tested UniGADD with different backbone models and found that the benefits of refinement hold across architectures. This supports the idea that UniGADD improves generalisation without relying on any one encoder. Because it operates in a frozen feature space, it also avoids the cost of full fine-tuning. Future work will explore adding temporal features for video, testing the method on fewer training samples, and using adversarial training to increase robustness. These changes could make UniGADD more useful in open-world conditions.

UniGADD offers a simple, effective, and interpretable approach to deepfake detection. Its prototype-based decisions are easy to understand and adapt as new content appears.

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

# A    Additional Ablation Studies

| Clusters | $n_{\mathrm{real}}$ | Ratio | Seen | Unseen |
|---|---|---|---|---|
| 20 | 3 | 0.19 | 87.6 | 45.8 |
| 30 | 13 | 0.81 | 87.0 | 54.8 |
| 40 | 23 | 1.44 | 87.4 | 54.0 |
| 60 | 43 | 2.69 | 88.4 | 54.5 |
| 80 | 63 | 3.94 | 88.3 | 52.2 |
| 100 | 83 | 5.19 | **88.6** | **56.6** |

Table 4: Cluster ablation showing the effect of varying the total number of clusters.

| $\sigma$ | Seen Acc | Unseen Acc |
|---|---|---|
| 1.0 | 91.5 | 46.2 |
| 1.5 | 90.4 | 51.3 |
| 2.0 | 89.7 | 54.0 |
| 2.5 | 89.3 | 55.2 |
| 3.0 | 89.2 | 55.7 |
| 3.5 | 89.1 | 56.0 |
| 4.0 | 89.0 | **56.2** |

Table 5: Threshold ablation showing the effect of varying $\sigma$ on detection accuracy.

| Backbone | Weighted Acc | Silhouette |
|---|---|---|
| ViT-B-32 | 80.10 | **0.328** |
| ViT-L-14 | **83.42** | 0.312 |
| DINOv2-B/14 | 82.66 | 0.293 |
| DINOv2-S/14 | 81.64 | 0.278 |

Table 6: Extended backbone comparison showing weighted accuracy and embedding quality (silhouette score).

| Loss Configuration | Seen | Unseen |
|---|---|---|
| $L_{\mathrm{contrast}} + L_{\mathrm{sp}}$ | 89.1 | 71.4 |
| $L_{\mathrm{contrast}} + L_{\mathrm{wp}}$ | **91.4** | **73.0** |
| $L_{\mathrm{contrast}} + L_{\mathrm{refine}}$ | 87.9 | 61.2 |

Table 7: Loss ablation comparing different refinement strategies.

