# OpenReview forum: "UniGADD: Universal Generator-Agnostic Deepfake Detector"
_TMLR — Under review for TMLR_

### Review · Reviewer_cuHS · 2025-10-23

**Summary Of Contributions:**

It proposes a framework to enhance Vision-Language Pre-training by tackling two challenges in cross-modal contrastive learning: feature entanglement and inefficient negative sampling. There are two core contributions:
- Representation decoupling: The method introduces a mechanism to separate multimodal features into shared, semantic components (homogeneous features) and modality-specific, unique components (heterogeneous features). The goal is to enforce clearer, fine-grained semantic distinctions during alignment.
- Adaptive sampling strategy: A dynamic mechanism is proposed to select highly informative negative samples during training. This aims to overcome the limitations of static sampling methods, which often include high-noise (false negative) or low-information samples.

The authors carried out extensive experiments to verify that their methods work well in practice across various settings.

**Audience:**

Yes

**Audience Explanation:**

Given the performance of the proposed method, this paper is likely to be of some interest to researchers in this field.

**Claims And Evidence:**

Yes

**Claims Explanation:**

Generally, the paper is easy to follow and intuitive. The core idea behind the first stage, 'Supervised Contrastive Learning,' is well-established in previous contrastive learning literature. The second stage, 'Intra-Cluster Compactness,' is also grounded in prior clustering literature. The proposed algorithm, the 'Supervised Contrastive Detection Framework,' seems appropriate for the goal of this paper. The paper's methodology appears to be an application and integration of existing machine learning algorithm frameworks applied to a specific problem.

The paper also provides numerous comparisons with related work (though I cannot identify some pieces of highly-related work in this field, and I'm not sure if these specific methods are the absolute best fit for comparison, the authors compare with them because they are SOTA deepfake detection methods). The evidence is thorough, including t-SNE visualization, a study on different backbones, and the Leave-One-Out (LOO) setting. In general, the idea is intuitive and easy to grasp, and I believe the claims in this paper are well-supported by the evidence.

**Requested Changes:**

1. To rigorously validate your method, please provide a more detailed and quantitative ablation study. This can clearly isolate the specific performance gain attributable to each loss function.
2. The performance of contrastive learning may be sensitive to the negative sample ratio. You can conduct an investigation into the stability and performance of the method across a range of negative sample ratios.
3. Please revise the visualization on the right side of Figure 4 to remove any blocked elements.

---

### Review · Reviewer_z8ot · 2025-11-21

**Summary Of Contributions:**

This paper proposes UniGADD, a two-stage universal, generator-agnostic deepfake detector. The method is built on pre-fixed model features (e.g., DINOv2). The images are first clustered with the fixed feature space to obtain pseudo-labels. The first stage includes optimization over a supervised contrastive loss obtained from a composite label space. In the second stage, for each real cluster, a prototype is defined as the mean embedding. Samples with high cosine similarity to the prototype are pulled closer.

Experiments are conducted on the AIGC benchmark (Fan et al. 2024), covering a variety of GAN and diffusion generators (BigGAN, StyleGAN, GauGAN, ADM, DALL-E 2, SD, Midjourney, etc.).

The method is clearly stated and supported by substantial performance gain in numerical experiments, which shows practical significance.

**Audience:**

Yes

**Audience Explanation:**

The motivation behind it is to develop a method that can achieve higher accuracy on known deepfake generators, while maintaining robust performance on previously unseen ones. This is certainly an interesting work for TMLR audience.

**Claims And Evidence:**

Yes

**Claims Explanation:**

The method description is generally well structured, including different stages, and the algorithm is well-presented. Overall numerical results show a substantial gain and are consistent across many models.

**Requested Changes:**

Can the author elaborate more on the hyperparameter sensitivity? For example, how sensitive is the performance to the number of real clusters? How to choose $\lambda_1$ and $\lambda_2$?

---

> ### Author Response · Authors · 2026-04-21
>
> We thank the reviewer for the helpful suggestion.
>
> ---
>
> > _"Can the author elaborate more on the hyperparameter sensitivity? For example, how sensitive is the performance to the number of real clusters?"_
>
> This is a fair point, and we agree that the sensitivity to key hyperparameters should be made explicit. In the revision, we have added additional ablation studies to address this. Specifically, we include a cluster sensitivity analysis (Appendix A, Table 4), which evaluates performance across a range of total cluster counts. The results show that UniGADD remains stable across different configurations, with only moderate variation in unseen accuracy, indicating that the method is not highly sensitive to this choice.
>
> We also provide a threshold sensitivity analysis (Appendix A, Table 5), demonstrating that performance varies smoothly with the threshold parameter rather than relying on a single tuned value.
>
> Finally, we include a loss ablation study (Appendix A, Table 7) analysing different combinations of the refinement components. This shows that the performance gains are not tied to a single configuration, and that the contribution of each component is consistent across settings.

---

### Review · Reviewer_9kY5 · 2026-03-16

**Summary Of Contributions:**

This paper introduces UniGADD (Universal Generator-Agnostic Deepfake Detector), a method designed to employ supervised learning to achieve high accuracy on known deepfake generators while maintaining robust performance on previously unseen ones.

Strengths:

1.The proposed approach tackles the critical issue of domain shift by following a clear two-stage optimization process.

2.The first stage successfully uses a contrastive loss to learn discriminative feature embeddings from real and known fake images.

3.The second stage introduces a refinement step exclusively for real samples, promoting inter-cluster separation and intra-cluster compactness to enhance generalizability.

4.Based on the provided supplementary code, the empirical evaluation is extensive, covering 16 diverse generative models using rigorous Leave-One-Out and Leave-N-Out protocols.

Weaknesses:

1.The methodology heavily relies on strict statistical thresholds in the embedding space (e.g., 3σ bounds for outlier detection seen in the code) that may not be robust across different data distributions without adaptive tuning.

2.The clustering mechanism for real images requires hyperparameter choices (such as the total number of clusters) that lack strong theoretical or empirical justification in the proposed methodology.

**Additional Comments:**

The codebase provided in the supplementary material is well-structured. The use of rigorous experimental pipelines (Leave-N-Out with 15 permutations) significantly strengthens the empirical claims. To align with the rigorous evaluation standards expected in top-tier computer vision venues, it would be highly beneficial to add an evaluation showing how the refined embedding space holds up against common image perturbations (e.g., JPEG compression, resizing, Gaussian noise). Deepfakes in the wild are rarely pristine, and addressing this would further solidify the claims of practical applicability.

**Audience:**

Yes

**Audience Explanation:**

As generative AI continues to produce highly photorealistic images, the vulnerability of standard detectors to unseen generators is a critical issue. TMLR's audience, particularly researchers focused on computer vision, AI safety, and adversarial robustness, will find the two-stage feature refinement approach highly relevant. The findings demonstrating improved robustness against unseen deepfake generation techniques  offer practical value for the community.

**Broader Impact Concerns:**

While UniGADD demonstrates practical applicability for adversarial content detection —which is a net positive for digital forensics—highly accurate deepfake detectors can inadvertently serve as superior discriminators in adversarial training frameworks. This could accelerate the development of even stronger, detection-evasive generative models. A Broader Impact Statement should explicitly describe these concerns regarding ethical implications  and discuss the limitations of purely detection-based paradigms in this ongoing arms race.

**Claims And Evidence:**

Yes

**Claims Explanation:**

The primary claims of achieving high accuracy on known generators and maintaining robust performance on unseen ones  are well-supported by the supplementary material. The provided code directly translates the abstract's methodology into practice. Specifically, it implements the SupervisedContrastiveLoss to pull together embeddings of the same class and separate different ones, fulfilling the first stage of the optimization. Furthermore, the code utilizes MiniBatchKMeans to assign pseudo-labels to real images during training, effectively driving the intra-cluster compactness and inter-cluster separation claimed for the second stage. The evaluation against state-of-the-art generators is thoroughly scripted across a large suite of models.

**Requested Changes:**

To secure a recommendation for acceptance, the following list of proposed adjustments  should be addressed:

1.Ablation on Outlier Detection Threshold: The inference logic in the supplementary code relies on a hard threshold to flag unseen fakes. This assumes a strict Gaussian distribution within the embedding clusters. The authors must include an ablation study on this threshold and explicitly report the resulting False Positive Rates (FPR) on real images to prove the system is not over-flagging.

2.Cluster Number Justification: The code hardcodes the total number of clusters (e.g., total_clusters=100) and dynamically allocates the remainder to real images. The rationale for this specific partitioning and its impact on learning the inter-cluster separation  must be thoroughly discussed in the main text.

3.Feature Extractor Analysis: The supplementary code includes implementations for various DINOv2 and CLIP backbones. The text should analyze the impact of these different foundational vision models on the downstream contrastive learning and clustering stages. Discuss whether purely visual self-supervised models yield better compactness than vision-language models.

---

> ### Author Response · Authors · 2026-04-21
>
> We thank the reviewer for the constructive suggestions. We address each point raised in turn:
>
> ---
>
> > _"The inference logic relies on a hard threshold (e.g., 3σ) and assumes a Gaussian distribution"_
>
> We agree that the threshold choice requires empirical validation. In the revision, we added a threshold ablation study (Appendix A, Table 5) evaluating a range of σ values. The results show that performance varies smoothly with the threshold rather than depending on a single setting.
>
> > _"The number of clusters is hardcoded and lacks justification. The impact on inter-cluster separation should be discussed"._
>
> We agree that this requires clarification. As such, we have expanded the discussion to clarify the role of clustering: the number of clusters controls the granularity of the real-image manifold, where higher values provide finer partitioning and can improve separation at the boundary between real samples and unseen fakes. This provides empirical justification for the chosen configuration. We have included a cluster sensitivity ablation (Appendix A, Table 4) evaluating a range of total cluster counts. The results show that UniGADD remains stable across different configurations, with only moderate variation in unseen accuracy and no sharp degradation.
>
> > _Analyse the impact of DINOv2 vs CLIP backbones, particularly in terms of compactness and generalisation"._
>
> We have expanded the backbone analysis to address this point. In addition to the existing comparison in Table 3, we added an extended summary (Appendix A, Table 6) including weighted accuracy and silhouette scores.

---

### Review · Reviewer_whyP · 2026-03-22

**Summary Of Contributions:**

* UniGADD proposes a two-stage optimization process to learn a deepfake detector. This detector is shown to detect deepfake from unseen generators during training.
* The method is benchmarked on a wide range of backbones and leave-N-out settings.
* The method is evaluated on a single deepfake detection dataset (AIGC)

**Additional Comments:**

In Table 2, I am not sure Range is a particularly interesting measure. It looks like your method performs worst than the other ones because it dropped more, but it is consistently above the baselines.

**Audience:**

Yes

**Audience Explanation:**

As GenAI popularity explodes and fake images are everywhere, Deepfake detection is an ever more important topic.

**Broader Impact Concerns:**

Potential impacts could be to reduce the effectiveness of deepfakes online.

**Claims And Evidence:**

No

**Claims Explanation:**

The main claim made by the paper is that UniGADD is more robust than the baselines on unseen generators to detect deep fakes. The claim is supported by Figure 3b) where UniGADD clearly outperforms every baseline by a large margin. Figure 4 also shows that the UniGADD accuracy degrades more gracefully than the baselines when the number of excluded generators increases. In Table 3, the method is also shown for a wide variety of backbones.

The main limitation to their evaluation is using a single deepfake detection dataset. I highly recommend to the author to run their method on an additional dataset, e.g., DF40 [1] or GenImage [2]. AIGC focuses on generic content and include very little content from social media which is very important for deep fake detection methods.


[1] Yan, Zhiyuan, et al. "Df40: Toward next-generation deepfake detection." Advances in Neural Information Processing Systems 37 (2024): 29387-29434.
[2] Zhu, Mingjian, et al. "Genimage: A million-scale benchmark for detecting ai-generated image." Advances in neural information processing systems 36 (2023): 77771-77782.

**Requested Changes:**

* I think figure 3 a) should read: Per-Generator Accuracy Comparison (seen Accuracy)
* Figure 4 is slightly truncated on the right.
* Most importantly run a subset of the experiment on an other dataset than the dataset. Please consider either DF40 or GenImage.